# Characterization of Vegetable Oils for Direct Use in Polyurethane-Based Adhesives: Physicochemical and Compatibility Assessment

**DOI:** 10.3390/ma18050918

**Published:** 2025-02-20

**Authors:** Żaneta Ciastowicz, Renata Pamuła, Łukasz Bobak, Andrzej Białowiec

**Affiliations:** 1Department of Applied Bioeconomy, Wrocław University of Environmental and Life Sciences, 37a Chełmońskiego Str., 51-630 Wrocław, Poland; 2Selena Industrial Technologies Sp. z o.o., Pieszycka 3, 58-200 Dzierżoniów, Poland; 3Department of Functional Food Products Development, Wrocław University of Environmental and Life Sciences, 51-630 Wrocław, Poland; lukasz.bobak@upwr.edu.pl

**Keywords:** bio-renewable oils, sustainable materials, carbon footprint, waste, polyurethane adhesives, decarbonization

## Abstract

This study evaluates the compatibility and innovative applications of unmodified vegetable oils, including rapeseed, sunflower, linseed, castor, and used cooking oils, in the production of sustainable polymeric materials, particularly polyurethane adhesives. Fatty acid composition was characterized using GC-MS, functional groups were identified by FTIR, and physicochemical properties, such as hydroxyl value, acid value, viscosity, and density, were measured using conventional analytical techniques. The results highlight significant differences in the properties of the oils, influencing their suitability for specific industrial applications. Castor oil, with its high ricinoleic acid content and hydroxyl value, was identified as the most suitable option for bio-based polyols and polyurethane production. Compatibility tests confirmed that unmodified oils can be effectively blended with polyols, ensuring stability and homogeneity without chemical modification. This approach simplifies production, reduces reliance on petrochemical feedstocks, and advances the development of environmentally friendly polyurethane adhesives. Future research will focus on optimizing formulations and assessing the long-term performance of adhesives incorporating unmodified vegetable oils.

## 1. Introduction

The shift from petrochemical feedstocks to renewable sources is crucial for sustainable development. Over the years, vegetable oils and their derivatives have been used in the production of biodiesel, lubricants, surfactants, and polymeric materials [1,2]. The benefits of using vegetable oils include their renewability, non-toxicity, widespread availability, and biodegradability, making them a viable alternative to traditional petrochemicals. Accurate knowledge of the physical and chemical properties of vegetable oils is crucial for ensuring controlled production, handling, and storage processes. To evaluate the quality of vegetable oils, it is essential to understand their physical properties, particularly their mechanical, rheological, and thermophysical characteristics.

Vegetable oils are essential commodities in the European Union (EU) and globally, serving as important agricultural products with significant roles in food production, biofuel manufacturing, and various industrial applications. In 2021, global vegetable oil production exceeded 200 million tons, with palm oil (41.3%), soybean oil (28.1%), and rapeseed oil (12.4%) being the dominant types. In the EU, vegetable oil production is dominated by rapeseed oil (49.9%), sunflower oil (18.4%), and olive oil (12.6%). The EU is one of the largest producers of vegetable oils, which are essential for both food and industrial applications. Although the EU does not produce significant quantities of palm oil or soybean oil, it relies on imports to meet domestic demand. Poland is a significant producer of vegetable oils within the EU, with a strong focus on rapeseed oil (92.2%). Other vegetable oils, such as sunflower oil (5.0%) and linseed oil (1.4%), are produced to a much lesser extent in Poland [3,4,5].

Castor oil is also present in the European market, although its production in the EU is minimal. Europe is one of the major importers of castor oil, sourcing it from countries like India, Brazil, Malaysia, Mexico, and China to meet the growing demand in the cosmetics, pharmaceutical, biofuel, and bioplastic production industries [4].

The shift from petrochemical feedstocks to renewable sources, like vegetable oils, marks a significant stride toward sustainable development. This transition brings numerous environmental benefits, encompassing aspects such as renewability, biodegradability, reduced greenhouse gas emissions, and lower toxicity.

The renewable and biodegradable nature of vegetable oils stems from their plant-based origin. In contrast to fossil fuels, which require millions of years to form, vegetable oils can be harvested on an annual basis. The renewability of vegetable oils ensures a sustainable supply without depleting natural resources, thereby supporting a more balanced and sustainable ecosystem. Vegetable oils are biodegradable, meaning they can be broken down by microorganisms into harmless substances. This property reduces the risk of long-term environmental contamination compared to petrochemical products, which can persist in the environment for decades. The biodegradability of vegetable oils is a significant advantage, as it lessens their impact on soil and water quality, thereby promoting healthier ecosystems [6]. The production and utilization of vegetable oils result in a reduction in greenhouse gas emissions when compared to those derived from petrochemical sources. During the growth phase, plants absorb carbon dioxide from the atmosphere, thereby partially offsetting the emissions generated during the extraction and processing of oil. This natural absorption helps to mitigate the overall carbon footprint. In contrast, the extraction and utilization of fossil fuels release significant amounts of carbon dioxide, contributing to climate change. The net carbon impact of vegetable oils is thus considerably lower, making them a more climate-friendly option [2,7]. Vegetable oils are typically non-toxic and present a relatively low risk to human health and the environment. They do not contain the same range of harmful chemicals or heavy metals that are often found in petrochemical products. This makes them a safer option for use in a variety of applications, including food, cosmetics, and pharmaceuticals. The lower toxicity of vegetable oils reduces the risk of environmental pollution and health hazards, contributing to safer and cleaner surroundings [8].

The utilization of vegetable oils is an effective strategy for promoting resource efficiency. Agricultural residues and by-products derived from oil extraction can be employed as animal feed or organic fertilizers, thereby contributing to a circular economy. This approach optimizes the utility of agricultural outputs, reduces waste, and enhances overall resource use efficiency. Furthermore, innovations in agricultural practices and crop genetics continue to enhance the yield and quality of oil-producing plants, thereby contributing to greater sustainability in agricultural and industrial processes. The application of advanced agricultural methods ensures that the cultivation of oil-producing plants is more productive and less resource intensive, thereby supporting long-term sustainability [9,10]. Moreover, the recycling of waste cooking oils (WCO) constitutes a pivotal element in the promotion of resource efficiency. Waste cooking oils can be repurposed for the production of biofuels and chemical feedstocks, thereby reducing dependence on fossil fuels and minimizing environmental pollution. By integrating waste cooking oils into the resource cycle, waste is further reduced, and the sustainability of various agricultural and industrial processes is enhanced. This comprehensive approach to utilizing both fresh vegetable oils and waste cooking oils exemplifies how recycling and resource efficiency can be effectively achieved, fostering a more sustainable future [11]. Additionally, the use of WCO for producing biolubricants and biosurfactants provides further environmental benefits by replacing virgin vegetable oils in industrial applications, thereby supporting a more circular economy [11,12].

Therefore, the objective of this study is to undertake a comparative analysis of the properties of selected vegetable oils that are critical for polymeric sustainable material production. The oils selected for this study are rapeseed oil, sunflower oil, linseed oil, castor oil, and used cooking oil. These oils were selected based on their prevalence and economic importance in the region. Castor oil, although less common, has many industrial applications. Waste cooking oil plays a key role in recycling and sustainability efforts within the EU. Understanding the detailed chemical composition of these oils is essential to optimize their use in industrial applications in line with EU sustainability and resource efficiency objectives.

Different analytical techniques were used to gain a full understanding of the chemical composition of studied vegetable oils. Gas chromatography-mass spectrometry (GC-MS) and Fourier transform infrared spectroscopy (FTIR) are two essential methods used to characterize and analyze these oils. These techniques provide detailed insights into the molecular structures, functional groups, and overall composition, which are essential for optimizing industrial applications [13,14]. The analyses were focused on the physicochemical properties of the oils, including their fatty acid composition, acid value, hydroxyl value, iodine value, peroxide value, water content, viscosity, and density.

## 2. Materials and Methods

### 2.1. Materials

Vegetable oils (rapeseed, sunflower, linseed, and castor) were purchased from local distributors Brenntag (Kędzierzyn-Koźle, Poland), Oquema (Ozorków, Poland), and Standard (Lublin, Poland). Spent cooking oils were obtained from Euro Eko Poland (Mielec, Poland) and Eneris Bioproten (Warsaw, Poland).

### 2.2. Analytical Methods

#### 2.2.1. Gas Chromatography

The oil samples were converted into the corresponding methyl esters according to the EN ISO 12966 method for the preparation of methyl esters of fatty acids, using boron trifluoride. The fatty acid profile was analyzed by using a gas chromatograph (GC6890) coupled with a mass spectrometer 5983 MS (Agilent Technologies Inc., Santa Clara, CA, USA) equipped with a quadrupole mass detector. Separation was performed in a capillary column HP-88 (0.25 mm × 100 m) filled with an 88:12 cyanopropyl-aryl poly-siloxane bed with a grain size of 0.2 μm. Helium was used as the mobile phase, and the sample was injected in the split mode at 4:1. The spectra were identified using the algorithm of searching the National Institute of Standards and Technology (NIST) library (2008 version) [15].

#### 2.2.2. Spectroscopy Analysis

The ATR-FTIR spectra were obtained using a Perkin-Elmer Spectrum Two spectrometer (Waltham, MA, USA) with a single reflection UATR accessory. The spectral range was 4000–450 cm^−1^ with a resolution of 4 cm^−1^, 4 scans, and 3551 data points.

#### 2.2.3. Acid Value (AV)

The acid number was determined by titration with potassium hydroxide solution according to EN ISO 660:2021-03. The sample was dissolved in a mixture of ethanol and diethyl ether using phenolphthalein as an indicator [16].

#### 2.2.4. Hydroxyl Value (OHV)

The hydroxyl value was determined following the ASTM D4274:2023 standard, Method E. The sample was subjected to a reaction with pyromellitic dianhydride in N,N-dimethylformamide (DMF), and the resulting acid was titrated with a sodium hydroxide solution utilizing an alcoholic solution of thymolphthalein as an indicator [17].

#### 2.2.5. Iodine Value (IV)

The iodine number was determined by titration with sodium thiosulfate solution according to EN ISO 3961:2018. The sample was dissolved in a mixture of cyclohexane and glacial acetic acid solvents, and starch was used as an indicator [18].

#### 2.2.6. Peroxide Value (PV)

The peroxide value was determined by titration with sodium thiosulfate solution according to EN ISO 3960:2017. The sample was dissolved in a mixture of glacial acetic acid and isooctane using starch as an indicator [19].

#### 2.2.7. Water Content

The water content was determined using the Karl Fischer titration method according to EN 12937:2000 using a Mettler Toledo C20 Series (Mettler Toledo, Greifensee, Switzerland) and Hydranal Coulomat AG (Honeywell, Seelze, Germany) [20].

#### 2.2.8. Elemental Analysis

The analysis of the elemental composition (C, H, N) was carried out following EN ISO 16948:2015 standards using a Perkin Elmer 2400 Series (Waltham, MA, USA). The O content was determined by mass balance (O% = 100 − C% − H% − N% − ash%) [21].

#### 2.2.9. Viscosity

Viscosity was measured using the Brookfield CAP 2000+ following EN ISO 3219-2:2021 (AMETEK Brookfield, Middleboro, MA, USA), utilizing a cone-plate rheometer setup with spindle S01 and 20 °C temperature [22].

#### 2.2.10. Density

Density was measured using a metal pycnometer following EN ISO 2811-1:2023 (POL-ZAF, Wroclaw, Poland) by filling the calibrated device with the test liquid, weighing it, and calculating the density based on the weight difference under controlled temperature conditions at 20 °C [23].

#### 2.2.11. Evaluation of Compatibility

Compatibility tests were conducted by mixing polyols and unmodified vegetable oils in a 50:50 weight ratio. The mixtures were stored at 23 °C for two weeks and 50 °C for 24 h to assess stability. Visual inspections were performed to evaluate homogeneity and detect phase separation or precipitates. Homogeneous mixtures were deemed compatible. This approach is commonly used in the evaluation of phase stability in polyol–polyol or polyol–oil systems.

## 3. Result and Discussion

### 3.1. Fatty Acid Composition of Plant Oils

The principal components of plant oils are triglycerides, which are esters formed from glycerol and fatty acids. Fatty acids make up approximately 95% of the total weight of triglycerides, and their composition is specific to each vegetable oil [24,25].

The chemical structures of the most popular fatty acids in vegetable oils are shown in Table 1.

The chemical structure of triglycerides, characterized by ester bonds and double bonds, provides a foundation for a wide range of industrial applications. From biodiesel production and polymer synthesis to the formulation of surfactants and lubricants, triglyceride modification is central to the development of sustainable and versatile industrial products [24]. The fatty acid percentage composition of tested plant oils is shown in Table 2.

The fatty acid composition of plant oils varies significantly, impacting their chemical properties and potential applications. This variability has been widely documented in the literature [25,26]. Oleic acid (C18:1) is present in varying concentrations among different oils, ranging from 5.68% in castor oil to 63.24% in used cooking oil [27]. Rapeseed oil also has a high oleic acid content (62.22%), making it one of the most abundant sources of this monounsaturated fat [28]. In contrast, sunflower oil contains a lower amount, with 30.93%. Linoleic acid (C18:2), a polyunsaturated fatty acid, is most concentrated in sunflower oil, where it reaches 58.48%, making it the most linoleic-rich oil in the dataset [29]. Other oils, such as linseed oil (15.06%) and rapeseed oil (14.7%), contain moderate amounts, while used cooking oil and castor oil have relatively lower levels at 13.36% and 5.94%, respectively. Alpha-linolenic acid (C18:3), another polyunsaturated fatty acid, is found at the highest concentration in linseed oil, where it constitutes 53.25% of the total fatty acids. Rapeseed oil contains 12.87%, while the presence of this fatty acid is minimal in sunflower (0.21%) and castor oil (0.43%). Palmitic acid (C16:0), a saturated fatty acid, is most prevalent in used cooking oil (12.59%), followed by rapeseed oil (5.71%) and sunflower oil (5.41%). Castor oil has the lowest palmitic acid content at 1.54%. Ricinoleic acid (C18:1 OH), a monounsaturated fatty acid with a hydroxyl group, is almost exclusively found in castor oil, where it constitutes 84.76%, making it a unique source of this fatty acid used in industrial applications [30,31]. The presence of other fatty acids, such as stearic acid (C18:0) and eicosenoic acid (C20:1), also varies across the oils but at much lower concentrations. In summary, the oils vary considerably in their fatty acid profiles, with oleic and linoleic acids being dominant in most oils, while linseed oil stands out for its high alpha-linolenic acid content and castor oil for its high ricinoleic acid content [32].

### 3.2. Fourier Transform Infrared Analysis of Plant Oils

Fourier transform infrared (FTIR) spectroscopy with attenuated total reflectance (ATR) is an advanced analytical technique utilized to investigate the molecular composition and structural features of plant oils. The FTIR-ATR spectra of various plant oils, including rapeseed oil, sunflower oil, linseed oil, castor oil, and used cooking oil, are shown in Figure 1.

The FTIR-ATR spectra provide crucial insights into the molecular composition and functional group characteristics of the analyzed vegetable oils. The interpretation of key absorption bands allows for a comprehensive understanding of the chemical structure and properties of these oils [14]. The broad absorption band observed around 3380–3400 cm^−1^ is indicative of hydroxyl (O-H) groups, particularly pronounced in castor oil due to its high hydroxyl content, distinguishing it from other vegetable oils that predominantly contain ester and alkyl groups [30,31]. The band appearing in the range of 3000–3020 cm^−1^ corresponds to the stretching vibrations of =CH- groups, typically found in unsaturated fatty acids, and is more prominent in oils rich in unsaturated fatty acids, such as linseed oil and sunflower oil. The bands at 2923 cm^−1^ and 2853 cm^−1^ are associated with the asymmetric and symmetric stretching of C-H bonds in methylene (CH_2_) groups, characteristic of the long hydrocarbon chains in fatty acids common to all the analyzed vegetable oils. The prominent peak at 1742 cm^−1^ is characteristic of the carbonyl (C=O) stretching vibration in ester groups, confirming the presence of triglycerides as the primary components across all oils [14]. In the range of 1645–1660 cm^−1^, the absorption is indicative of C=C stretching vibrations from unsaturated fatty acids, which is crucial for assessing the degree of unsaturation within the oils, with higher intensity seen in more unsaturated oils, like linseed oil [33]. The absorption at 1462 cm^−1^ corresponds to the bending vibrations of methylene (CH_2_) groups, reflecting the structural characteristics of the alkyl chains in fatty acids, while the band at 1377 cm^−1^ is associated with the bending vibrations of methyl (CH_3_) groups, indicating the presence of terminal methyl groups in the fatty acid chains. Bands within the ranges of 1220–1240 cm^−1^, 1150–1170 cm^−1^, and 1096–1110 cm^−1^ are related to C-O stretching vibrations in ester groups, consistent across all analyzed oils, highlighting the common esterified structure of triglycerides [14]. Finally, the absorption at 722 cm^−1^, due to the rocking motion of CH_2_ groups in long alkyl chains, further confirms the presence of extended hydrocarbon chains typical of fatty acids in vegetable oils [34]. The FTIR-ATR spectral analysis thus provides a detailed molecular fingerprint of the vegetable oils, enabling the identification of functional groups, evaluation of unsaturation levels, and detection of any structural modifications due to oxidation or thermal degradation [35]. These insights are critical for the quality control, authenticity verification, and suitability assessment of vegetable oils in various industrial applications. The high hydroxyl content in castor oil makes it a superior candidate for polyurethane production due to its reactive O-H groups. In contrast, oils rich in unsaturated fatty acids, like linseed oil, may pose challenges due to their higher susceptibility to oxidation, potentially affecting the stability of polyurethane products.

### 3.3. Chemical and Physical Characteristics of Vegetable Oils

Vegetable oils exhibit diverse chemical and physical properties that are crucial for their application in various industries. The following section presents a comparison of five selected vegetable oils—rapeseed oil, sunflower oil, linseed oil, castor oil, and used cooking oil—focusing on key physicochemical parameters, such as acid value, hydroxyl value, iodine value, peroxide value, water content, viscosity, and density (Table 3). These parameters not only define the quality of the oils but also their suitability for different industrial processes.

The acid value is an indicator of free fatty acids in oils, which reflects their level of hydrolytic degradation. Rapeseed oil, with the lowest AV of 0.15 mg KOH/g, is the most stable oil in this respect, while used cooking oil, with the highest AV of 0.94 mg KOH/g, demonstrates significant degradation due to prolonged use. This aligns with the general expectation that fresh oils have lower AVs, indicating better quality, while used oils, due to breakdown of triglycerides, exhibit higher acid values [24,36].

The hydroxyl value indicates the amount of hydroxyl groups in an oil, which is important for applications in the production of polyurethanes and resins. Vegetable oils generally have a low hydroxyl number because they predominantly contain non-hydroxylated fatty acids. However, the hydroxyl value of used cooking oil can vary widely, depending on the degree of oxidation and the type of oil originally used. Typically, it is higher than in fresh oils due to the formation of hydroxylated oxidation products during cooking [37,38].

Castor oil is notable for its exceptionally high OHV (158.04 mg KOH/g), attributed to the presence of ricinoleic acid, making it highly suitable for chemical processes that require cross-linking reactions. In contrast, rapeseed oil (0.66 mg KOH/g) and sunflower oil (2.92 mg KOH/g) have much lower hydroxyl values, indicating that they are less suited for applications involving cross-linking reactions.

The iodine value indicates the degree of unsaturation in oils. Linseed oil has the highest IV (173.2 g I_2_/100 g), indicating a high degree of unsaturation and strong drying properties, which make it ideal for applications in coatings, paints, and varnishes. In contrast, castor oil, with an IV of 85.5 g I_2_/100 g, has lower unsaturation, making it less reactive in oxidative environments [2,25].

The peroxide value is a measure of primary oxidation in oils. Fresh oils tend to have lower PVs, with rapeseed oil at 1.85 meq O_2_/kg, reflecting high oxidative stability. In contrast, used cooking oil exhibits the highest PV (32.8 meq O_2_/kg), suggesting significant oxidation and reduced quality. Linseed oil and castor oil also show higher PVs (7.65 meq O_2_/kg and 16.75 meq O_2_/kg, respectively), limiting their use in applications where oxidative stability is a critical factor [24,39].

The water content directly affects the quality and stability of oils. Rapeseed oil and sunflower oil, with water contents of 0.04% and 0.05%, respectively, exhibit the best stability, while used cooking oil (0.20%) shows higher water content, reflecting the degradation and contamination commonly associated with repeated use. Castor oil, with 0.15% water, also poses challenges for moisture-sensitive applications [40,41].

The viscosity of oils impacts their flow properties and usability in industrial applications, such as biofuels and lubricants. Castor oil, with an exceptionally high viscosity of 900 mPas, is suited for applications requiring thick, viscous oils. In contrast, sunflower oil (63.2 mPas) and linseed oil (55.5 mPas) have much lower viscosities, making them easier to handle and process. Used cooking oil, with a viscosity of 102.7 mPas, presents challenges due to the presence of degradation products, which increase its viscosity. The density of oils plays a role in their transport, storage, and flow characteristics. Most vegetable oils fall within the range of 0.9–0.95 g/cm^3^ [24,42]. Castor oil, with the highest density of 0.955 g/cm^3^, is heavier and denser than the other oils, making it less suitable for applications requiring lightweight oils. On the other hand, rapeseed oil (0.916 g/cm^3^) and sunflower oil (0.910 g/cm^3^) have lower densities, making them more versatile for a wide range of industrial uses.

In conclusion, castor oil stands out for its exceptionally high hydroxyl value and viscosity, making it highly suitable for specialized chemical processes, such as polyurethane and resin production. Rapeseed oil, with its low acid value, oxidative stability, and low water content, is versatile and well-suited for a wide range of industrial applications. Both linseed oil and used cooking oil present significant challenges. Linseed oil’s high iodine and peroxide values limit its oxidative stability, while the extensive degradation of used cooking oil makes it less suitable for applications requiring long-term stability and high-quality performance.

### 3.4. Results of Elemental Analysis

The elemental composition of vegetable oils is essential for understanding their potential uses, especially in the production of composite materials, like polyurethanes. The carbon (C), hydrogen (H), nitrogen (N), and oxygen (O) content, typically determined through CHN analysis, reveals critical insights into the reactivity and stability of these oils in industrial formulations [43,44].

In this study, five vegetable oils—rapeseed oil, sunflower oil, linseed oil, castor oil, and used cooking oil—were analyzed for their elemental composition. The results are summarized in Table 4.

The elemental composition data, as presented in Table 4, show remarkable differences between the selected vegetable oils. The carbon content varied from 73.36% in sunflower oil to 78.33% in used cooking oil. The hydrogen content ranged from 11.54% in linseed oil to 13.10% in used cooking oil. The nitrogen content, although relatively low in all samples, was highest in used cooking oil (1.66%) and lowest in castor oil (0.76%). The oxygen content was inversely related to the carbon content, with the highest oxygen content in linseed oil (15.58%) and the lowest in used cooking oil (6.91%). The high carbon and hydrogen content in the used cooking oil suggests that it has undergone significant changes due to thermal degradation and the accumulation of carbon-rich compounds during cooking. This could potentially make it a more reactive additive in polyurethane composites, offering increased cross-link density and improved mechanical properties. However, the higher nitrogen content could lead to undesirable side reactions or affect the stability of the resulting composites. Rapeseed oil, with its moderate carbon and hydrogen content and low nitrogen content, has a balanced profile, making it a suitable candidate for applications requiring consistent and predictable behavior. Its oxygen content indicates the presence of functional groups that could facilitate compatibility and bonding with polyurethane matrices. Sunflower and linseed oils have similar carbon and hydrogen content but differ significantly in their oxygen content. The higher oxygen content in linseed oil could be attributed to the presence of more unsaturated fatty acids, which could also pose challenges in terms of oxidative stability. Castor oil, known for its high ricinoleic acid content, has a lower carbon content than the other oils, together with a moderate hydrogen content and the lowest nitrogen content. Despite its relatively low total oxygen content, castor oil contains hydroxyl groups due to the ricinoleic acid structure. These hydroxyl groups increase its reactivity with isocyanates, making it highly valued in polyurethane production. This property is advantageous in applications where efficient curing and strong interfacial bonding are required.

The comparative analysis of the physical and chemical properties of various vegetable oils highlights significant variations that influence their industrial applications. This study confirms that the selection of vegetable oil for specific applications must be based on a thorough understanding of these properties. The fatty acid profiles of the oils demonstrate significant diversity. For instance, rapeseed oil’s high oleic acid content makes it suitable for applications requiring oxidative stability. In contrast, sunflower oil and linseed oil are highly polyunsaturated, making them preferable for applications that benefit from such properties. Castor oil, with its unique ricinoleic acid content, is valuable for producing bio-based polyols, while used cooking oil, due to its varied composition, poses challenges in consistency but holds potential for recycling efforts. The study’s detailed analysis of acid, hydroxyl, iodine, peroxide values, water content, viscosity, and density reveals critical insights. The acid value (AV) indicates the quality and degradation level of oils. Fresh oils, like rapeseed and sunflower, have low AVs, making them suitable for high-quality applications, whereas high AV in used cooking oil signifies significant degradation. The hydroxyl value (OHV) is critical for applications involving polyurethane production. Castor oil, with the highest OHV, is ideal for such uses due to its hydroxylated fatty acids. The iodine value (IV) reflects the degree of unsaturation, with linseed oil’s high IV suggesting strong drying properties, suitable for paints and coatings. The peroxide value (PV) measures primary oxidation, affecting oil stability. The lower PV in rapeseed and sunflower oils indicates better stability compared to linseed and castor oils. The water content impacts the stability and suitability for specific industrial processes; fresh oils exhibit a lower water content, while used cooking oil shows higher levels, indicating potential issues in storage and handling. Viscosity and density are crucial factors in the processing and application of oils. However, the oils tested exhibited minimal variation within the established ranges.

### 3.5. Evaluation of Compatibility Between Unmodified Oils and Polyurethane Components

Compatibility tests were carried out to assess whether unmodified vegetable oils, such as rapeseed and castor oils, can be effectively incorporated into typical polyurethane components without prior chemical modification.

Samples of polyols were blended with unmodified vegetable oils in a 50:50 weight ratio and the stability of the blends was evaluated under controlled conditions. The blends were stored at 23 °C for 2 weeks and at 50 °C for 24 h. Visual analysis was carried out to assess homogeneity and to detect any phase separation or precipitate formation.

The visual evaluation of the compatibility tests, presented in Figure 2, Figure 3 and Figure 4, illustrates the stability and homogeneity of the oil–polyol mixtures under the tested conditions.

The visual evaluation confirms that unmodified vegetable oils can be effectively blended with polyols without chemical modification, maintaining homogeneity under the conditions tested. The vegetable oil/polyol blends demonstrated stability at both 23 °C for 2 weeks and 50 °C for 24 h, with no significant phase separation or precipitate formation observed. These results highlight the feasibility of directly using unmodified vegetable oils in polyurethane formulations, offering a simpler and more sustainable alternative to traditional chemically modified oils.

### 3.6. Innovative Application of Unmodified Vegetable Oils in Polyurethane Adhesives

In contemporary polyurethane adhesive production, petrochemical-derived raw materials dominate, contributing significantly to the carbon footprint. With increasing global emphasis on sustainable development and greenhouse gas reduction, renewable resources, such as vegetable oils, have attracted substantial interest. Most current studies focus on the chemical transformation of vegetable oils into biopolyols or bioplasticizers, a process that necessitates energy-intensive, multi-step procedures.

Our approach emphasizes the direct use of unmodified vegetable oils as liquid additives or fillers in polyurethane formulations. This innovative application simplifies the production process, reduces costs, and significantly lowers the carbon footprint by replacing petrochemical components with renewable alternatives. Unmodified oils, such as rapeseed, sunflower, and castor oils, offer dual functionality: improving the flexibility of the adhesives while reducing their environmental impact.

This approach opens new possibilities for reducing dependency on petrochemical feedstocks and advancing the development of environmentally friendly polyurethane adhesives. Further investigations will focus on optimizing oil-to-formulation ratios and assessing the long-term stability and performance of adhesives incorporating unmodified vegetable oils.

## 4. Conclusions

The study demonstrated that vegetable oils can be effectively used in sustainable polyurethane production, reducing reliance on petrochemical feedstocks. Rapeseed oil exhibits high oxidative stability, making it a valuable raw material for various industrial applications. Sunflower and linseed oils, rich in polyunsaturated fatty acids, show potential for the production of environmentally friendly coatings; however, their susceptibility to oxidation requires further evaluation of the stability of the final product. Castor oil, due to its high hydroxyl value, is the most suitable candidate for bio-based polyol synthesis. Used cooking oil, despite its variable composition, aligns with the principles of a circular economy and offers opportunities for industrial recycling.

Compatibility tests indicate that unmodified vegetable oils can directly be incorporated into polyurethane formulations. Further research should focus on optimizing the formulation to ensure that the oil content does not negatively impact the mechanical strength of polyurethane adhesives. Additionally, mechanical stability should be evaluated under accelerated aging conditions with variable temperature and humidity to determine the durability of these materials over time.

## Figures and Tables

**Figure 1 materials-18-00918-f001:**
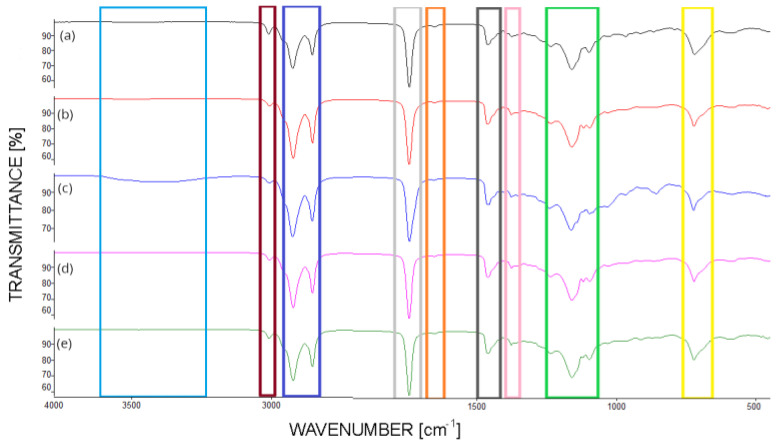
FTIR spectra of (**a**) linseed oil; (**b**) used cooking oil; (**c**) castor oil; (**d**) rapeseed oil; (**e**) sunflower oil.

**Figure 2 materials-18-00918-f002:**
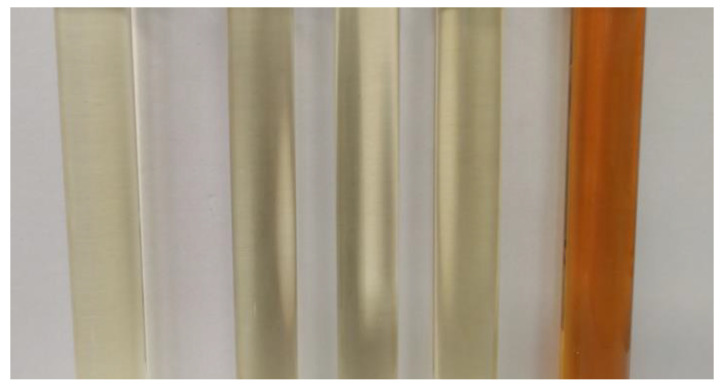
Initial appearance of samples, with oils from left to right: rapeseed, sunflower, linseed, castor, and used cooking oil.

**Figure 3 materials-18-00918-f003:**
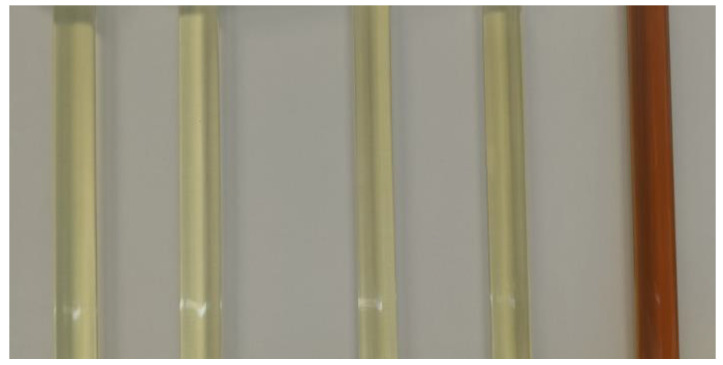
Samples after 2 weeks at 23 °C from left to right: rapeseed, sunflower, linseed, castor, and used cooking oil.

**Figure 4 materials-18-00918-f004:**
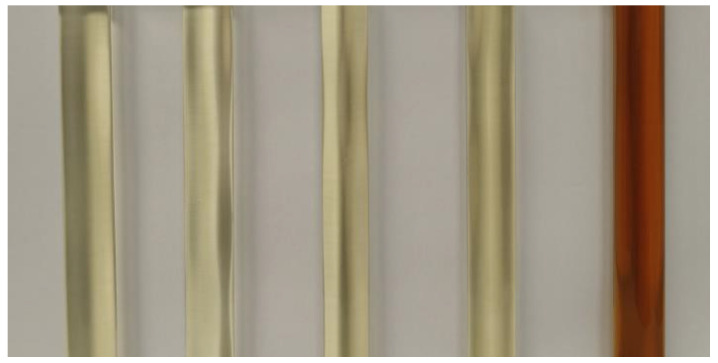
Samples after 24 h at 50 °C from left to right: rapeseed, sunflower, linseed, castor, and used cooking oil.

**Table 1 materials-18-00918-t001:** Fatty acids in plant oils. (SFA) = Saturated fatty acid, (MUFA) = Monounsaturated fatty acid, (PUFA) = Polyunsaturated fatty acid. Compiled from literature data [24,25].

Name Acid	Symbol	Type	Formula
Palmitic	C16:0	SFA	CH_3_(CH_2_)_14_COOH
Palmitoleic	C16:1	MUFA	CH_3_(CH_2_)_5_CH=CH(CH_2_)_7_COOH
Stearic	C18:0	SFA	CH_3_(CH_2_)_16_COOH
Oleic	C18:1	MUFA	CH_3_(CH_2_)_7_CH=CH(CH_2_)_7_COOH
Linoleic	C18:2	PUFA	CH_3_(CH_2_)_4_(CH=CHCH_2_)_2_(CH_2_)_2_COOH
Linolenic (α and γ)	C18:3	PUFA	CH_3_CH_2_(CH=CHCH_2_)_3_(CH_2_)_6_COOH CH_3_(CH_2_)_4_(CH=CHCH_2_)_3_(CH_2_)_2_COOH
Ricinoleic	C18:1 OH	MUFA with OH	CH_3_(CH_2_)_5_CH(OH)CH_2_CH=CH(CH_2_)_7_COOH
Arachidic	C20:0	SFA	CH_3_(CH_2_)_18_COOH
Eicosenoic	C20:1	MUFA	CH_3_(CH_2_)_7_CH=CH(CH_2_)_9_COOH
Eicosadienoic	C20:2	PUFA	CH_3_(CH_2_)_4_(CH=CHCH_2_)_2_(CH_2_)_8_COOH
Behenic	C22:0	SFA	CH_3_(CH_2_)_20_COOH
Erucic	C22:1	MUFA	CH_3_(CH_2_)_7_CH=CH(CH_2_)_11_COOH

**Table 2 materials-18-00918-t002:** Fatty acids in plant oils.

	Vegetable Oils				
Fatty Acid [%]	Rapeseed Oil	Sunflower Oil	Linseed Oil	Castor Oil	Used Cooking Oil
C16:0	5.71	5.41	5.55	1.54	12.59
C16:1	0.77	0.64	0.13	-	0.4
C18:0	1.76	4.33	3.98	1.65	3.52
C18:1	62.22	30.93	22.03	5.68	63.24
C18:2	14.7	58.48	15.06	5.94	13.36
C18:3	12.87	0.21	53.25	0.43	6.89
C18:1 OH	-	-	-	84.76	-
C20:0	0.77	-	-	-	-
C20:1	1.09	-	-	-	-

**Table 3 materials-18-00918-t003:** Physicochemical properties of vegetable oils.

	Rapeseed Oil	Sunflower Oil	Linseed Oil	Castor Oil	Used Cooking Oil
Properties	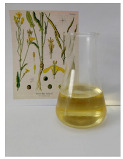	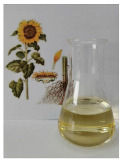	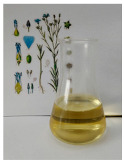	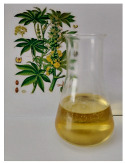	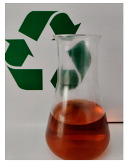
Acid Value (AV) [mgKOH/g]	0.15	0.21	0.41	0.91	0.94
Hydroxyl Value (OHV) [mgKOH/g]	0.66	2.92	3.73	158.04	4.22
Iodine Value (IV) [gJ_2_/100 g]	110.6	124.1	173.2	85.5	98.0
Peroxide Value (PV) [meqO_2_/kg]	1.85	5.18	7.65	16.75	32.8
Water Content [%]	0.04	0.05	0.06	0.15	0.20
Viscosity [mPas]	71.8	63.2	55.5	900	102.7
Density [g/cm^3^]	0.916	0.910	0.928	0.955	0.918

**Table 4 materials-18-00918-t004:** Elemental composition of selected vegetable oils.

	Rapeseed Oil	Sunflower Oil	Linseed Oil	Castor Oil	Used Cooking Oil
Elemental [wt.%]					
C	75.92	73.6	76.73	74.00	78.33
H	11.96	12.23	11.54	12.70	13.10
N	0.96	1.09	1.15	0.76	1.66
O	11.16	13.32	15.58	12.54	6.91

## Data Availability

The original contributions presented in this study are included in the article. Further inquiries can be directed to the corresponding authors.

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
