# Peer review of "Characterization of Vegetable Oils for Direct Use in Polyurethane-Based Adhesives: Physicochemical and Compatibility Assessment"

_materials, 2025, doi:10.3390/ma18050918_

Round 1
Reviewer 1 Report
Comments and Suggestions for Authors
The submitted article conducts an investigation on the use of various vegetable oils in the production of polymeric materials. The approach adopted to explore sustainable and biodegradable sources as replacements for petrochemical raw materials is relevant in the current context of searching for ecological alternatives.
Linguistic Corrections: There are some language errors that need correction, the use of tenses and prepositions presents inconsistencies that can compromise the clarity of the text. It is recommended that the manuscript undergo a careful review by a native English speaker or a professional specialized in editing scientific texts in English.
The study presents important contributions to the field of material science and has the potential to positively impact the development of new sustainable technologies.
Comments on the Quality of English Language
There are English language errors that need to be addressed, especially in the use of tenses and prepositions, which can impact the clarity of the text. It is recommended that the manuscript undergo a thorough review by a native speaker or an expert in scientific English.
Reviewer 2 Report
Comments and Suggestions for Authors
Reviewer comments
- A brief summary: this paper intitled “Comparative Analysis of Vegetable Oils: Properties and Potential in Polymeric Material Production” with strength contribution which aims to evaluates the compatibility and innovative applications of unmodified vegetable oils, including rapeseed, sunflower, linseed, castor, and used cooking oils, in the production of sustainable polymeric materials, particularly polyurethane adhesives.
- Specific comments
o Abstract: this sentences is wrong. “….Analytical techniques such as GC-MS and FTIR spectroscopy were employed to characterize key physicochemical properties, including fatty acid composition, hydroxyl value, viscosity, and density”
o Hydroxyl value, viscosity, and density are not determined with the GC and FTIR.
o Introduction : Figure 1. Is blurred. Please replace it. (You can remove it. It is not needed in the introduction)
o Materials and methods: give references of each method.
o Line 110 and 128: unify the writing of (spent used cooking oil) using (used cooking oil).
o Give more information about the norms used such as: (EN ISO 660: year), (ASTM D4274-…)……
o Result and discussion section:
o Figure 2. Triglyceride structure is already known. Remove it.
o Some references and citations are missed in the discussion.
o “summary” section should be replaced by “conclusion” and needs to be reduced.
o Clearly define the key findings and implications rather than just summarizing methods.
o Instead of general statements like "Future research will optimize formulations", specify what variables need optimization.
o The cited references are not recent and more than 50% was before 5 years. The format of the references is not accorded with the journal instruction. They therefore need to be revised.
Reviewer 3 Report
Comments and Suggestions for Authors
The manuscript entitled “Comparative Analysis of Vegetable Oils: Properties and Potential in Polymeric Material Production” is a manuscript related with the description of several properties of several vegetable oils and their potential industrial use. The introduction needs to be re-written, the inclusion of the main objective at the end of the section could be helping to clarify the manuscript. Material and methods need to be re-written; discussion needs to be improved, the conclusions needs to be re-written.
Comments
Title
Please re-write title, vegetable oil is general, also polymeric material is general too.
Introduction
Line 48 – 49 Please add statistical data about imports in EU
Results and discussion
Table 1 was made with experimental data of from literature review, please clarify it
Table 2
Line 202 – 204 The fatty acid profile of the vegetable oils is known to show high variability, please use proper cites to discuss it
Line 217 – 219 Castor oil ia widely used in several industries, please add proper cites
Conclusions
Please re-write the section
The main drawback of the manuscript, is the novelty
Reviewer 4 Report
Comments and Suggestions for Authors
The manuscript evaluates the compatibility and applications of unmodified vegetable oils (rapeseed, sunflower, linseed, castor and used cooking oils) in the production of polyurethane adhesives to reduce dependence on petrochemical raw materials. Analytical techniques were used (GC-MS and FTIR spectroscopy) to characterize physicochemical properties of oils and products. The analytical methods used for characterization are standard analytical methods (EN ISO 12966, EN ISO 660, ASTM D4274 standard, Method 150 E, EN ISO 3961...), however a description of these methods and limitations and interferences during the analyses is necessary for a better understanding of the work. I strongly recommend an improvement in the description of methods and technical details.
Although the text is well structured, some suggestions could be made to improve the overall text before publication.
· Pag 2, line 14. Use appropriate abbreviation to European Union as mentioned before on page 1, line 39. Check in all manuscript text.
· Pag 4. What is the confidence level (%) of the comparison data performed by the database (NIST) for the compounds? Which parameters and restrictions were used for comparison. The comparison spectra/data should be added in the supplementary material.
· Pag 7. Lines 227-229. The sentence: “This method provides detailed information about 227 functional groups and molecular interactions by measuring the absorbance of infrared 228 radiation at different wavelengths.” It is a redundant sentence and should be removed from the text.
· Pag 7, Figure 3. In the regions between 4000-3750 and in the regions of 2700-1900 cm-1, the absorption spectrum in the IR region must be cut for better visualization and comparison of the absorption bands, since they are "clean" regions of absorption of functional groups.
· Section. 3.2. Fourier Transform Infrared Analysis of plant oils There are several papers in the literature that analyze the vibration modes in the infrared region for fatty acids and such articles should be cited, since the band assignments made by the authors are not referenced.
· Few references to 2023 and 2024 are observed. Improve discussions and insert some recent references.
· There is a branch of problems with references. Check and standardize the formatting of references. For instance: [27] Elektroenergetyka_nr_06_06_e1.
After general considerations, the manuscript can be forwarded for publication in materials Journal.
Round 2
Reviewer 2 Report
Comments and Suggestions for Authors
Good luck
Reviewer 3 Report
Comments and Suggestions for Authors
The authors included all suggestions from reviewers or answered the questions, thus, in my opinion, the revised manuscript is ready to be accept